# Imitation from Arbitrary Experience: A Dual Unification of Reinforcement and Imitation Learning Methods

**Harshit Sikchi**
The University of Texas at Austin
hsikchi@utexas.edu

**Amy Zhang**
The University of Texas at Austin
amy.zhang@austin.utexas.edu

**Scott Niekum**
University of Massachusetts Amherst
sniekum@cs.umass.edu

## Abstract

It is well known that Reinforcement Learning (RL) can be formulated as a convex program with linear constraints. The dual form of this formulation is unconstrained, which we refer to as dual RL, and can leverage preexisting tools from convex optimization to improve the learning performance of RL agents. We show that several state-of-the-art deep RL algorithms (in online, offline, and imitation settings) can be viewed as dual RL approaches in a unified framework. This unification calls for the methods to be studied on common ground, so as to identify the components that actually contribute to the success of these methods. Our unification also reveals that prior off-policy imitation learning methods in the dual space are based on an unrealistic coverage assumption and are restricted to matching a particular $f$-divergence. We propose a new method using a simple modification to the dual framework that allows for imitation learning with arbitrary off-policy data to obtain near-expert performance.

## 1 Introduction

Most of the modern deep Reinforcement Learning (RL) algorithms are built on the foundations of the approximate dynamic programming (ADP) formulation [2]. This formulation led to a family of temporal difference algorithms and to some of the most performant RL algorithms. Model-free deep RL algorithms such as SAC [17], TD3 [12], DDPG [36] and model-based deep RL algorithms such as Dreamer [18], TD-MPC [19], LOOP [48], MuZero [44] and MBPO [24] exemplify this class of successful algorithms. These methods have accomplished various feats including solving challenging games such as Starcraft, Chess, Go, and Shogi while also being very successful in robotics for learning manipulation and locomotion policies.

In this work, we consider an alternate formulation for RL, which views RL as a convex objective with linear constraints. This framework has been formalized before [40] and has been known in its linear programming form since the work of [38, 5]. The convex program for RL can be converted into a dual unconstrained form that admits the tools already present in convex optimization literature, making it more suitable for stochastic optimization. The methods arising from the convex program formulation of RL also have the benefit of being truly off-policy: they can use arbitrary off-policy data to better estimate the on-policy policy gradient, i.e they can implicitly perform distribution correction. For this reason, the methods in this space have often been referred to as DICE (DIstribution Correction Estimation) methods in previous literature [41, 31, 35, 37, 56].

Reincarnating Reinforcement Learning Workshop at ICLR 2023

Our first contribution is a demonstration that several recent algorithms in deep RL and imitation learning can be viewed as dual RL methods in a unified framework. These algorithms represent state-of-the-art methods [15, 57, 34, 14] in reinforcement and imitation learning, for both online and offline settings. A number of these recent works have utilized differing tools to derive their method (e.g Gumbel regression [15], change of variables [14, 57], lower bounded Q-function [34]) which makes it difficult to study them on common ground. We hope that our presented unification provides a framework for future methods to perform evaluation and analyze which factors actually make the algorithm better or worse.

Second, building upon the dual framework, we propose a new algorithm for off-policy imitation learning that is able to leverage arbitrary off-policy data to learn near-expert policies, relaxing the coverage assumption of previous works [37, 7, 57, 29]. Our resulting algorithm, ReCOIL, is simple, non-adversarial, and admits a single-player optimization in contrast to previous works in imitation [16, 21, 10, 47]. We empirically demonstrate the failure of previous imitation learning methods based on the coverage assumption in the dual setting. We also evaluate our methods for learning to imitate in offline continuous control settings on a set of MuJoCo environments and show competitive performance. Finally, we point out scope for future exploration in designing better algorithms for deep RL and imitation learning methods utilizing the dual framework.

**The Dual-RL Landscape**

Figure 1: We show that a number of prior methods are special cases of the dual RL framework. Based on this framework, we also propose new methods addressing the shortcomings of previous works (boxed in green).

## 2 Preliminaries

A Markov Decision Process (MDP) is defined by the tuple $(\mathcal{S}, \mathcal{A}, p, r, d_0)$ with state-space $\mathcal{S}$, action-space $\mathcal{A}$, transition probability $p(s_{t+1}|s_t, a_t)$, reward function $r(s, a)$, and initial state distribution $d_0(s)$. In the infinite horizon discounted MDP, the goal of reinforcement learning algorithms is to maximize the return for policy $\pi$ given by $J^\pi = \mathbb{E}_{a_t \sim \pi(s_t), s_0 \sim d_0}[\sum_{t=0}^{\infty} \gamma^t r(s_t, a_t)]$.

**Value functions:** $V^\pi : \mathcal{S} \to \mathbb{R}$ represents a state-value function which estimates the return from the current state $s_t$ and following policy $\pi$, defined as $V^\pi(s) = \mathbb{E}_{a_t \sim \pi(s_t)}[\sum_{t=0}^{\infty} \gamma^t r(s_t, a_t)|s_0 = s]$. Similarly, $Q^\pi : \mathcal{S} \times \mathcal{A} \to \mathbb{R}$ represents an action-value function, usually referred as a Q-function, defined as $Q^\pi(s, a) = \mathbb{E}_{a_t \sim \pi(s_t)}[\sum_{t=0}^{\infty} \gamma^t r(s_t, a_t)|s_0 = s, a_0 = a]$. Value functions corresponding to the optimal policy $\pi^*$ are defined to be $V^*$ and $Q^*$. The value function can be updated by minimizing mean-squared error with the target given by Bellman operator $\mathcal{T}^{\pi_Q}$ in the ADP setting:

$$\mathcal{T}^{\pi_Q} Q(s_t, a_t) = r(s_t, a_t) + \mathbb{E}_{s_{t+1} \sim p, a_{t+1} \sim \pi_Q}[\gamma(Q(s_{t+1}, a_{t+1})], \tag{1}$$

where $\pi_Q$ is updated to be greedy with respect to $Q$, the current Q-function. We will use $\mathcal{T}V(s, a)$ to denote $r(s, a) + \mathbb{E}_{s_{t+1} \sim p(\cdot|s_T, a_t)}[V(s_{t+1})]$. Note the absence of an explicit policy in this operator.

**State-action/State visitation distribution:** The state-action visitation distribution (stationary occupancies) $d^\pi(s, a) : \mathcal{S} \times \mathcal{A} \to [0, \infty)$ of $\pi$ is:

$$d^\pi(s, a) = (1 - \gamma) \sum_{t=0}^{\infty} \gamma^t P(s_t = s, a_t = a|s_0 \sim \rho_0, a_t \sim \pi(s_t), s_{t+1} \sim p(s_t, a_t)). \tag{2}$$

The state-visitation distribution marginalizes over actions on $d^\pi(s, a)$ and will be overloaded by the notation $d^\pi(s) = \sum_{a \in \mathcal{A}} d^\pi(s, a)$. In this work, we will use $d^O$, $d^R$, and $d^E$ to denote offline, replay-buffer, and expert state-action visitation distribution respectively.

$f$-**divergence:** Let $f : (0, \infty) \to \mathbb{R}$ be a convex lower semi-continuous function with $f(1) = 0$. Let $P$ and $Q$ be two probability distributions, then the $f$-divergence is defined as:

$$D_{\mathrm{f}}(P \,||\, Q) = \mathbb{E}_{z \sim Q}\left[ f\left(\frac{P(z)}{Q(z)}\right)\right].\qquad(3)$$

## 3   A Dual Unification for Reinforcement Learning Methods

A number of recently proposed algorithms for deep reinforcement learning and imitation learning use different mathematical tools to derive their method (e.g Gumbel regression [15], change of variables [14, 57], lower-bounded Q-function [34]) which makes it difficult to study them on common ground. This paper aims to connect a number of algorithmic developments in deep reinforcement learning as a special case in a simple unified framework of dual reinforcement learning. (True) off-policy algorithms, that can leverage arbitrary off-policy data for policy improvement, are a simple by-product of this previously known but mostly ignored in-practice framework [40] we consider. In Section 3.1, we first describe the general framework for dual reinforcement learning and then proceed to chalk out the connections for reinforcement learning and imitation learning in Section 3.2 and Section 3.3 respectively. In Section 4, we proceed to propose a new method, `ReCOIL`, for imitation learning from arbitrary experience.

### 3.1   Dual Reinforcement Learning

We will consider the regularized policy optimization setting in this work given below:

$$\max_{\pi} \mathbb{E}_{d^\pi(s,a)}[r(s, a)] - \alpha D_{\mathrm{f}}(d^\pi(s, a) \,||\, d^O(s, a)),\qquad(4)$$

where $d^O$ is a known state-action visitation distribution, $\alpha$ is the temperature parameter that allows us to weigh policy improvement against conservatism by staying close to the state-action distribution $d^O$, and $f$ denotes a particular $f$-divergence. We can rewrite the problem as a convex optimization problem (CoP) by considering optimization over *valid* state-action visitations satisfying bellman-flow constraints:

$$\textbf{Q-CoP:}\quad \max_{\pi, d \geq 0} \mathbb{E}_{d(s,a)}[r(s, a)] - \alpha D_{\mathrm{f}}(d(s, a) \,||\, d^O(s, a))\qquad(5)$$

$$\text{s.t}\ \ d(s, a) = (1 - \gamma)d_0(s).\pi(a|s) + \gamma \sum_{s',a'} d(s', a')p(s|s', a')\pi(a|s),$$

The above problem is overconstrained – the inner maximization w.r.t $d$ is unnecessary as the $(\mathcal{S} \times \mathcal{A})$ constraints uniquely determine the distribution. We can relax the constraints to get an equivalent but different form for the problem:

$$\textbf{V-CoP:}\quad \max_{d \geq 0} \mathbb{E}_{d(s,a)}[r(s, a)] - \alpha D_{\mathrm{f}}(d(s, a) \,||\, d^O(s, a))\qquad(6)$$

$$\text{s.t}\ \ \sum_{a \in \mathcal{A}} d(s, a) = (1 - \gamma)d_0(s) + \gamma \sum_{s',a'} d(s', a')p(s|s', a').$$

We can leverage Lagrangian duality to convert the convex program above into its dual unconstrained optimization form (see Appendix A.1 for derivation and review) since the objective is convex and the constraints are linear.

In summary, we have two methods for policy optimization given by:

$$\texttt{dual-Q} : \max_{\pi} \min_{Q} (1 - \gamma)\mathbb{E}_{d_0(s),\pi(a|s)}[Q(s, a)] + \alpha \mathbb{E}_{s,a \sim d^O}[f^*\left(\left[\mathcal{T}^\pi Q(s, a) - Q(s, a)\right]/\alpha)\right)]$$
$$(7)$$

$$\texttt{dual-V} : \min_{V(s)} (1 - \gamma)\mathbb{E}_{d_0(s)}[V(s)] + \alpha \mathbb{E}_{s,a \sim d^O}[f^*\left(\left[\mathcal{T}V(s, a) - V(s)\right]/\alpha\right)].\qquad(8)$$

where $Q$ and $V$ are Lagrange variables in this framework and $f^*$ denotes the convex conjugate of function $f$. We overload $y(s,a)$ to denote $r(s,a) + \gamma \sum_{s'} p(s'|s,a)\pi(a'|s')Q(s',a') - Q(s,a)$ or $r(s,a) + \gamma \sum_{s'} p(s'|s,a)V(s') - V(s)$ for `dual-Q` and `dual-V` settings respectively for notational simplicity. We note an important property of the dual RL formulation: at convergence, the distribution ratio between the optimal policy and distribution used for regularization is captured in the first derivative of the conjugate $f$-divergence. Formally,

$$\frac{d^*(s,a)}{d^O(s,a)} = f^{*'}\left(\frac{y(s,a)}{\alpha}\right). \tag{9}$$

The dual formulation has two appealing properties: (a) The gradient of the objective (Eq 7 and 8) w.r.t $\pi$ as $Q$ is optimized is the on-policy policy gradient using off-policy data[1] (b) Allows us to incorporate regularization of different forms such as w.r.t behavior data (pessimism [53]), data generated by previous policy (trust region [26, 45, 46]), uniform distribution (max-entropy [17]) or an online planner [48]. In the dual formulation above, we have ignored the constraints that imply that the distribution $d(s,a)$ should be always positive $\forall\, s, a$. The closed-form solutions for the dual formulation under the positivity constraint can be found in Appendix A.2. The positivity constraints have the effect of changing the function $f^*$ in Equation 7 and 8 to $f_p^*$ (refer Appendix A.2).

### 3.2 Connections to duality in reinforcement learning

Equations 5 and 6 provide a natural choice for both offline and online reinforcement learning algorithms. Using the dual RL framework, we obtain a set of approaches that leverage `dual-Q` and `dual-V` for policy improvement. In this section, our first result shows that Conservative Q-learning [34], an offline RL method primarily understood to prevent overestimation by learning a lower bounded Q function is actually a `dual-Q` method. Second, we show that Extreme Q-Learning (X-QL) [15], a method for both online and offline RL based on the principle of *implicit maximization* in the value function space using Gumbel regression, can be reduced to a `dual-V` problem with a semi-gradient update rule (i.e `stop-gradient` $(r(s,a) + \gamma \sum_{s'} p(s'|s,a)V(s'))$) when $f$ is set to be the reverse-KL regularization. This insight obtained through duality, allows us to propose a class of algorithms extending X-QL, by choosing different functions $f$ which we show below to result in a family of implicit maximizers. We defer the detailed discussion on unifying RL methods using duality to Appendix A.3 with Lemmas 5, 6, 7 and focus on the imitation learning setting in the main paper due to space constraints.

### 3.3 Connections to duality in imitation learning

In this section, we will first discuss the setting of offline imitation learning with expert data only – where the agent has no access to the environment and is limited to a fixed amount of offline expert transitions. In this direction, we consolidate prior work and also propose a new algorithm arising from the dual-V formulation. Then, we discuss the off-policy imitation learning setting where the agent has access to limited expert data that it is trying to imitate along with some suboptimal data which is obtained offline or through agent online interactions. We discuss prior works from the perspective of duality and show how they are limited by their assumption of coverage (off-policy replay data covers expert data) or their reliance on a particular $f$-divergence. Finally, we propose a new method for off-policy imitation learning that relaxes the coverage assumption and works for arbitrary $f$-divergences.

#### 3.3.1 Offline imitation learning with expert data only

We start with the `dual-Q` and `dual-V` equations and repurpose them for imitation by simply setting the reward function to be uniformly 0 across the state-action space and setting the regularization distribution to be the expert distribution $d^O(s,a) = d^E(s,a)$. We use $\mathcal{T}_0^\pi$ and $\mathcal{T}_0$ to denote the backup operator with zero rewards. `dual-Q` (offline imitation) takes the following form:

`dual-Q` (offline imitation with expert-only data):

$$\max_\pi \min_Q (1-\gamma)\mathbb{E}_{d_0(s),\pi(a|s)}[Q(s,a)] + \mathbb{E}_{s,a\sim d^E}[f^*\left([\mathcal{T}_0^\pi Q(s,a) - Q(s,a)]/\alpha\right)]. \tag{10}$$

---

[1]Note that the on-policy policy gradient is with respect to the regularized Q-values

Interestingly, this reduction directly leads us to the equivalence of an imitation learning method IQLearn [14] derived using a change of variables in the form of an inverse backup operator. [14] uses this method in the online imitation learning setting with an additional regularization which we suggest is unprincipled, also pointed out by others [1] (as only expert data samples can be leveraged in the above optimization) and provide a fix in the Section 4.

**Lemma 1.** `dual-Q` *is equivalent to IQ-Learn when* $r(s, a) = 0 \ \forall \ (\mathcal{S}, \mathcal{A})$ *and* $d^O(s, a) = d^E(s, a)$.

Utilizing Lemma 1 above, we also provide a new observation in the form of Corollary 1 below. We find that Implicit Behavior Cloning [7], a method that performs behavior cloning using a contrastive objective, is a special case of `dual-Q` for offline imitation learning.

**Corollary 1.** `dual-Q` *is equivalent to Implicit Behavior Cloning [7] when* $r(s, a) = 0 \ \forall \ (\mathcal{S}, \mathcal{A})$ *and* $d^O(s, a) = d^E(s, a)$ *and* $f$ *is set to be the total variation divergence.*

Leveraging the `dual-V` variant we analogously obtain a new method for offline imitation learning, which we refer to as `IVLearn` that we leave for future work. (See Appendix A.4.1).

### 3.3.2 Off-policy imitation learning (under coverage assumption)

The `dual-Q` and `dual-V` framework does not naturally extend to off-policy imitation learning. To remedy this, prior methods have relied on careful selection of an $f$-divergence and a coverage assumption that allows them to arrive at an off-policy objective for imitation learning [57, 22, 37]. *Under the assumption that the replay buffer visitation (denoted by* $d^R$*) covers the expert visitation* ($d^R > 0$ *wherever* $d^E > 0$*)* [37], which we refer to as the **coverage assumption**, and $f$ set to be the reverse KL divergence, we get the following `dual-Q` and `dual-V` forms for imitation (see Appendix A.5 for derivation):

`dual-Q` for off-policy imitation (coverage assumption):

$$\max_{\pi(a|s)} \min_{Q(s,a)} (1 - \gamma)\mathbb{E}_{\rho_0(s), \pi(a|s)}[Q(s, a)] + \mathbb{E}_{s, a \sim d^R}[f^*(\mathcal{T}^\pi_{r^{imit}} Q(s, a) - Q(s, a))], \tag{11}$$

where $\mathcal{T}^\pi_{r^{imit}}$ denote backup operator under the reward function $r^{imit} = -\log \frac{d^R(s,a)}{d^E(s,a)}$. This choice of KL divergence leads us to a reduction of another method, OPOLO [57] for off-policy imitation learning to `dualQ` which we formalize in the lemma below:

**Lemma 2.** `dual-Q` *for off-policy imitation learning reduces to OPOLO [57], with the* $f$*-divergence set to the reverse KL divergence when* $r(s, a) = 0 \ \forall \mathcal{S}, \mathcal{A}, d^O = d^E$ *and under the assumption that the replay data distribution covers the expert data distribution.*

We note that the `dual-V` framework for off-policy imitation learning under coverage assumptions was studied in the imitation learning work SMODICE [37].

## 4 A Method for Imitation Learning from Arbitrary Experience

In this section, we propose a new approach to relax the coverage assumption discussed above for imitation as well as allow for arbitrary $f$-divergence matching in the dual framework. We refer to our method as `ReCOIL` (RElaxed Coverage for Off-policy Imitation Learning). We consider an alternate optimization objective that fits well in the dual RL framework by changing the regularization to be between mixture distributions. Concretely, we are interested in the following $f$-divergence regularization:

$$D_{\text{f}}(\beta d(s, a) + (1 - \beta)d^R(s, a) \| \beta d^E(s, a) + (1 - \beta)d^R(s, a)). \tag{12}$$

Minimizing this divergence alone is a valid imitation learning objective [16, 27, 43, 47] since the global minimum for this objective is achieved at $d = d^E$. Let $d^R_{mix} := \beta d(s, a) + (1 - \beta)d^R(s, a)$ and $d^{E,R}_{mix} := \beta d^E(s, a) + (1 - \beta)d^R(s, a)$. The modified Q-CoP for imitation learning objective is given by:

$$\max_{d(s,a) \geq 0, \pi(a|s)} -D_{\text{f}}(d^R_{mix}(s, a) \| d^{E,R}_{mix}(s, a))$$

$$\text{s.t } d(s, a) = (1 - \gamma)\rho_0(s).\pi(a|s) + \gamma\pi(a|s) \sum_{s',a'} d(s', a')p(s|s', a') \tag{13}$$

The V-CoP for this setting can be similarly specified and the dual results in two variants of ReCOIL that can be leveraged for off-policy imitation learning with arbitrary data as formalized by Lemma 3 and Lemma 4 below.

**Lemma 3.** *(`dual-Q` for off-policy imitation (relaxed coverage assumption)) Imitation learning using off-policy data can be solved by optimizing the following modified dual objective for Q-CoP with $r(s,a) = 0 \, \forall \mathcal{S}, \mathcal{A}$ and $f$-divergence considered between distributions $d_{mix}^R := \beta d(s,a) + (1 - \beta)d^R(s,a)$ and $d_{mix}^{E,R} := \beta d^E(s,a) + (1 - \beta)d^R(s,a)$, and is given by:*

$$\max_{\pi(a|s)} \min_{Q(s,a)} \beta(1-\gamma)\mathbb{E}_{d_0(s),\pi(a|s)}[Q(s,a)] + \mathbb{E}_{s,a\sim d_{mix}^{E,R}}\left[f_p^*(\mathcal{T}_0^\pi Q(s,a) - Q(s,a))\right]$$
$$- (1-\beta)\mathbb{E}_{s,a\sim d^R}[\mathcal{T}_0^\pi Q(s,a) - Q(s,a)] \tag{14}$$

Analogously, we have the following result for off-policy imitation learning in the V-space.

**Lemma 4.** *(`dual-V` for off-policy imitation (relaxed coverage assumption)) Imitation learning using off-policy data can be solved by optimizing the following modified dual objective for V-CoP with $r(s,a) = 0 \, \forall \mathcal{S}, \mathcal{A}$ and $f$-divergence considered between distributions $d_{mix}^R := \beta d(s,a) + (1 - \beta)d^R(s,a)$ and $d_{mix}^{E,R} := \beta d^E(s,a) + (1 - \beta)d^R(s,a)$, and is given by:*

$$\min_{V(s)} \beta(1-\gamma)\mathbb{E}_{d_0(s)}[V(s)] + \mathbb{E}_{s,a\sim d_{mix}^{E,R}}\left[f_p^*(\mathcal{T}_0 V(s,a) - V(s))\right]$$
$$- (1-\beta)\mathbb{E}_{s,a\sim d^R}[\mathcal{T}_0 V(s,a) - V(s)] \tag{15}$$

In the above-proposed methods, we avoid the pitfalls of previous methods based on coverage assumption that requires learning a reward function $\left(\log(\frac{d^R(s,a)}{d^E(s,a)})\right)$ that is ill-defined in state-action space with zero expert support. Our methods also implicitly learns a distribution ratio $\frac{\beta d(s,a)+(1-\beta)d^R(s,a)}{\beta d^E(s,a)+(1-\beta)d^R(s,a)}$ as $f^{*'}(y(s,a))$ which is well-defined for all replay buffer and expert transitions ($d^R > 0$ or $d^E > 0$) that the policy is trained on.

## 5 Experimental Analysis

Our experimental evaluation aims to answer the following questions: (a) Does the (true) off-policy nature of dual RL framework allow it to fix the brittleness in traditional off-policy methods? (b) Does ReCOIL allow for better estimation of agent visitation distribution? (c) How does ReCOIL compare to baselines for learning to imitate from offline datasets of mixed quality?

### 5.1 The failure of ADP-based traditional off-policy algorithms

Our experiments with the popular off-policy method SAC [17] reveal its brittleness to off-policy data. At the beginning of training, each learning agent is provided with expert or human-demonstrated trajectories for completing the task. We add 1000 transitions from this dataset to the replay buffer for the off-policy algorithm to bootstrap from. As the action dimension increases, the brittleness of SAC becomes more apparent (see SAC+off policy data and SACfD plots in Figure 3). We hypothesize that this failure in the online RL setting is primarily due to the training instabilities caused by TD-backups resulting in overestimation in regions where the agent's current policy does not visit. In Figure 4, we observe that overestimation indeed happens in environments with larger action dimensions and these overestimations take longer to get corrected and in the process destabilize the training. Figure 3 in Appendix D.1 shows that the `dual-Q` method for RL (AlgaeDICE) is able to leverage off-policy data to increase learning performance without any signs of destabilization. This can be attributed to the distribution correction estimation property of dual RL methods which updates the current policy using the corrected on-policy policy visitation [41]. Note, that we set the temperature $\alpha$ to a low value (0.001) to disentangle the effect of pessimism which is an alternate way to avoid overestimation.

### 5.2 Does `ReCOIL` allow for better estimation of agent visitation distribution?

We consider the proposed ReCOIL method and investigate its ability to estimate distribution ratios correctly using the property shown in Eq 9. We consider the inner optimization for ReCOIL-Q to obtain the distribution ratio for a fixed policy $\pi$. Thus, given the visitation distribution of the replay

buffer $d^R$, expert $d^E$, and the policy $\pi$, the inner optimization implicitly learn the distribution ratio between $d^R_{mix}$ and $d^{E,R}_{mix}$, allowing us to infer agent visitation $d^\pi$. We consider two settings in our experiments: (1) a 2-timestep MDP where the agent starts from state $s_0$ and transitions to one of the states $\{s_1, s_2, s_3, s_4, s_5\}$ which are absorbing. In this setting, the replay buffer perfectly covers the unknown ground truth agent visitation. (2) a 2-D gridworld where the agent can move cardinally and the replay buffer distribution does not cover the unknown ground truth agent visitation.

Our results (Figure 5 and 6 in Appendix D.2) demonstrate that in the perfect coverage setting, ReCOIL is able to infer the agent policy visitation perfectly, and in the case of imperfect coverage is able to significantly outperform other methods. Note that we modify SMODICE to incorporate state-action expert data rather than relying on state-only expert data. IQLearn does not leverage replay data information, relying only on expert data to infer agent visitation. SMODICE's reward function $-\log \frac{d^R(s,a)}{d^E(s,a)}$, arising from its coverage assumption is ill-defined in parts of state space where the expert has no support leading to poor downstream density ratio estimation.

### 5.3 Benchmarking performance of ReCOIL on MuJoCo tasks

| Dataset | Env | RCE | ORIL | SMODICE | ReCOIL |
|---|---|---|---|---|---|
| random+expert | hopper | 51.41±38.63 | 73.93±11.06 | **101.61±7.69** | 95.04±4.48 |
| | halfcheetah | 64.19±11.06 | 60.49±3.53 | 80.16±7.30 | **84.10±4.72** |
| | walker2d | 20.90±26.80 | 2.86±3.39 | **105.86±3.47** | 100.84±6.34 |
| | ant | 105.38±14.15 | 73.67±12.69 | **126.78±5.12** | 126.74±4.63 |
| random+few-expert | hopper | 25.31±18.97 | 42.04±13.76 | 60.11±18.28 | **79.44±13.53** |
| | halfcheetah | 2.99±1.07 | 2.84±5.52 | 2.28±0.62 | **3.90±0.66** |
| | walker2d | 40.49±26.52 | 3.22±3.29 | **107.18±1.87** | 83.23±19.00 |
| | ant | 67.62±15.81 | 25.41 ± 8.58 | -6.10±7.85 | **94.25± 8.30** |
| medium+expert | hopper | 29.37±3.39 | 61.35±7.91 | 54.77±6.2 | **63.26±4.99** |
| | halfcheetah | 61.14±18.31 | 57.15±3.49 | 58.01±3.12 | **84.01±3.40** |
| | walker2d | 19.84±2.71 | 4.14±2.69 | 1.2±1.57 | **91.51±16.50** |
| | ant | 81.18±29.73 | 103.42±8.43 | 102.82±4.63 | **119.94±4.52** |

Table 1: Normalized scores for ReCOIL on the different D4RL datasets along with 1000 given expert transitions compared to different offline IL baselines. ReCOIL improves over baselines when the offline dataset has low coverage of expert or when it is more challenging to distinguish expert vs suboptimal transitions.

In this task, we use an offline dataset of environment interactions from D4RL [8]. We consider the following MuJoCo environments: Hopper, Walker2d, HalfCheetah, and Ant. We consider three different dataset compositions for the offline dataset - 'random+expert', 'medium+expert', and 'random+few-expert'. The first two datasets consist of a few expert trajectories ($\leq$200) and a large number of suboptimal transitions (1 million). random+few-expert has only $\leq$ 30 expert trajectories. The learning agent also has access to 1000 expert transitions. We compare recent methods for offline imitation learning with suboptimal data – RCE [6], SMODICE [37] and ORIL [59]. We do not compare to DEMODICE [29] as SMODICE was shown to be competitive in [37].

We focus on ReCOIL-V due to its favorable property of being a single-player non-adversarial optimization. Note, that contrary to SMODICE [37] which is based on the coverage assumption and requires learning a discriminator, ReCOIL just learns a single parametric $V$ function. Table 1 empirically validates that ReCOIL-V outperforms or is competitive over baselines in all environments. SMODICE shows poor performance in cases when the dataset has low expert coverage (random+few-expert) or where the discriminator is not able to reliably distinguish expert vs non-expert data (medium+expert).

## 6 Related Work

**Developments in deep model-free reinforcement learning methods:** The promise of off-policy learning falls short as learning off-policy with function approximators presents a number of issues like over-estimation, training instability, and various biases [51, 9, 12, 33]. Previous works have approached fixing these issues using methods like doubleQ learning [20], target networks [39], emphatic weightings [25, 23] among other approaches. These approaches still do not carry over well to the offline RL setting where overestimation bias is more likely since unlike online RL, the policy cannot correct its overestimation by deploying the current policy. A number of solutions

exists for controlling overestimation in prior work – $f$-divergence regularization to the training distribution [53, 42, 13, 11], support regularization [49], implicit maximization [32], learning a Q function that penalizes OOD actions [34] and learning in a pessimistic MDP [28, 55]. A recent method XQL [15] uses implicit maximization to obtain significant gains in learning performance across online and offline RL settings. Prior work [54, 52, 32, 42] investigates specialized solutions to incorporate previously collected off-policy data during online learning. The dual RL framework [40] fixes the issue of distribution mismatch in ADP-based off-policy reinforcement learning works that ignore the distribution shift [17, 12] and provides a principled solution.

**Developments in deep model-free imitation learning methods:** Imitation learning has benefited greatly by using off-policy data to improve learning performance [30, 43, 47, 57]. Often, replacing the on-policy expectation common in most Inverse RL formulations [58, 50] by expectation under off-policy samples, which is unprincipled, has led to large gains in sample efficiency [30]. Principled offline imitation learning using expert data only is a simple by-product of the dual-RL framework. Previous works have proposed a solution in the dual RL space to this problem based on an unrealistic coverage assumption [37, 57, 29] and restricting themselves to matching a particular $f$-divergence. In this work, we relax this assumption and allow for generalizing to all $f$-divergences, presenting a principled off-policy approach to imitation. Our work also presents an approach that allows for single-player non-adversarial optimization for imitation learning in contrast to previous works [31].

## 7    Conclusion

Dual reinforcement learning algorithms have great potential for developing performant Deep RL methods. Indeed, we show that a number of the recently developed methods for RL – Extreme Q-learning, Conservative Q-learning, and Imitation learning – OPOLO, IQLearn, and Implicit Behavior Cloning can be unified through the lens of an already existing framework of dual RL. Our insight calls for these methods to be studied under this unified lens to figure out components that contribute to the success of these methods. As a by-product of this unification, we are able to identify novel insights that give us a way to propose: a family of implicit maximizers that avoids the overestimation problem due to actor-critic learning by relying on in-distribution maximization, an offline single-player non-adversarial offline imitation learning method from expert data only, and a general off-policy imitation learning method from arbitrary data that relaxes the restrictive coverage assumption made by prior work. We analyze the off-policy imitation learning method by experiments in continuous control and demonstrate superior performance.

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

# A  Appendix

## A.1  A Review of Dual-RL

In this section, we aim to give a self-contained review for Dual Reinforcement Learning. For a more thorough read, refer to [40].

### A.1.1  Convex conjugates and $f$-divergence

We first review the basics of duality in reinforcement learning. Let $f : (0, \infty) \to \mathbb{R}$ be a convex function. The convex conjugate $f^*$ of $f$ is defined by:

$$f^*(y) = \sup_{x \in \mathbb{R}} [\langle x, y \rangle - f(x)] \tag{16}$$

where $\langle \, \rangle$ denotes the dot product. The convex conjugates have the important property that $f^*$ is also convex and the convex conjugate of $f^*$ retrieves back the original function $f$. Going forward, we would be dealing extensively with $f$-divergences. Informally, $f$-divergences [] are a measure of distance between two probability distributions. Here's a more formal definition:

Let $f : (0, \infty) \to \mathbb{R}$ be a convex lower semi-continuous function with $f(1) = 0$. Let $P$ and $Q$ be two probability distributions, then the $f$-divergence is defined as:

$$D_{\mathrm{f}}(P \, || \, Q) = \mathbb{E}_{z \sim Q} \left[ f \left( \frac{P(z)}{Q(z)} \right) \right] \tag{17}$$

Now, we will do a simple exercise in finding the convex conjugate for this $f$-divergence (where $f$ is a convex function) which will also give us the well-known variational representation of $f$-divergence. We will use it frequently in the subsequent sections.

Using the definition of convex conjugate and the fact that convex conjugate of $f^*$ gives back $f$, we have:

$$D_{\mathrm{f}}(P \, || \, Q) = \mathbb{E}_{z \sim Q} \left[ f \left( \frac{P(z)}{Q(z)} \right) \right] \tag{18}$$

$$= \sup_{y} \mathbb{E}_{z \sim Q} \left[ \frac{P(z)}{Q(z)} y(z) \right] - \mathbb{E}_Q[f^*(y(z))] \tag{19}$$

$$= \sup_{y : \mathcal{Z} \to \mathbb{R}} \mathbb{E}_{z \sim P}[y(z)] - \mathbb{E}_{z \sim Q}[f^*(y(z))] \tag{20}$$

Thus Eq 20 derives the variational form for $f$-divergence. Although deriving the analytical form of $f^*$ is not complicated for most common $f$-divergences — set the derivative for Eq 16 to zero and find out the stationary point, it might be useful to list some common $f$-divergences and their conjugates $f^*$. We also note an important relation regarding $f$ and $f^*$: $(f^*)' = (f')^{-1}$, where the $'$ notation denotes first derivative.

Table 2: List of $f$-divergences and their convex conjugates

| Divergence | $f(t)$ | $f^*(u)$ |
|---|---|---|
| Forward KL | $-\log t$ | $-1 - \log(-u)$ |
| Reverse KL | $t \log t$ | $e^{(u-1)}$ |
| Squared Hellinger | $(\sqrt{t} - 1)^2$ | $\frac{u}{1-u}$ |
| Pearson $\chi^2$ | $(t-1)^2$ | $u + \frac{u^2}{4}$ |
| Total variation | $\frac{1}{2}|t - 1|$ | $u$ |
| Jensen-Shannon | $-(t+1) \log(\frac{t+1}{2}) + t \log t$ | $-\log(2 - e^u)$ |

### A.1.2  Duality in Reinforcement Learning

Duality in reinforcement learning allows a different perspective for solving RL problems, often giving off-policy alternatives to typical on-policy approaches. We consider a regularized policy optimization

objective below:

$$\max_{\pi} \mathbb{E}_{d^\pi(s,a)}[r(s,a)] - \alpha D_{\mathrm{f}}(d^\pi(s,a) \,||\, d^o(s,a)) \tag{21}$$

where $d^o$ is a known state-action visitation distribution. Optimizing over $\pi$, at first sight, gives us a non-convex problem further complicating the analysis. We can rewrite the problem as a linear program (LP) by considering optimization over *valid* state-action visitations by adding a constraint for the optimization:

$$\max_{\pi, d \geq 0} \mathbb{E}_{d(s,a)}[r(s,a)] - \alpha D_{\mathrm{f}}(d(s,a) \,||\, d^O(s,a)) \tag{22}$$

$$\text{s.t } d(s,a) = (1-\gamma)d_0(s).\pi(a|s) + \gamma \sum_{s',a'} d(s',a')p(s|s',a')\pi(a|s) \tag{23}$$

where $\alpha$ allows us to weigh policy improvement against conservatism from staying close to the state-action distribution $d^O$.

A careful reader may notice that the above problem is overconstrained. The solution to the inner maximization with respect to $d$ is uniquely determined by the $|\mathcal{S}| \times |\mathcal{A}|$ constraints from the formulation. The inner optimization, using only the constraints, uniquely determines the visitation $d^\pi$ - the state action visitation of policy $\pi$ and is independent of the term being optimized 22. The gradient with respect to policy $\pi$ when $d$ is optimized can be shown to be equivalent to the on-policy policy gradient (see Section 5.1 from [40]).

The constraints above are the probability flow equations that a stationary state-action distribution must satisfy. Now, how can we go about solving it? Here is where duality comes into play. First, we form the lagrangian dual of our original optimization problem, transforming our constrained optimization into an unconstrained form. This introduces additional optimization variables - the Lagrange multipliers.

$$\max_{\pi, d \geq 0} \min_{Q(s,a)} \mathbb{E}_{s,a \sim d(s,a)}[r(s,a)] - \alpha D_{\mathrm{f}}(d(s,a) \,||\, d^o(s,a))$$

$$+ \sum_{s,a} Q(s,a) \left( (1-\gamma)d_0(s).\pi(a|s) + \gamma \sum_{s',a'} d(s',a')p(s|s',a')\pi(a|s) - d(s,a) \right)$$

where $Q(s,a)$ are the Lagrange multipliers for enforcing the equality constraints. We can now do some algebraic manipulation on the above equation to further simplify it:

$$\max_{\pi, d \geq 0} \min_{Q(s,a)} \mathbb{E}_{s,a \sim d(s,a)}[r(s,a)] - \alpha D_{\mathrm{f}}(d(s,a) \,||\, d^O(s,a))$$

$$+ \sum_{s,a} Q(s,a) \left( (1-\gamma)d_0(s).\pi(a|s) + \gamma \sum_{s',a'} d(s',a')p(s|s',a')\pi(a|s) - d(s,a) \right) \tag{24}$$

$$= \max_{\pi, d \geq 0} \min_{Q(s,a)} (1-\gamma)\mathbb{E}_{d_0(s),\pi(a|s)}[Q(s,a)]$$

$$+ \mathbb{E}_{s,a \sim d}\left[ r(s,a) + \gamma \sum_{s'} p(s'|s,a)\pi(a'|s')Q(s',a') - Q(s,a) \right] - \alpha D_{\mathrm{f}}(d(s,a) \,||\, d^O(s,a)) \tag{25}$$

$$= \max_{\pi(a|s)} \min_{Q(s,a)} \max_{d(s,a) \geq 0} (1-\gamma)\mathbb{E}_{d_0(s),\pi(a|s)}[Q(s,a)]$$

$$+ \mathbb{E}_{s,a \sim d}\left[ r(s,a) + \gamma \sum_{s'} p(s'|s,a)\pi(a'|s')Q(s',a') - Q(s,a) \right] - \alpha D_{\mathrm{f}}(d(s,a) \,||\, d^O(s,a)) \tag{26}$$

$$= \max_{\pi(a|s)} \min_{Q(s,a)} \max_{d(s,a) \geq 0} \frac{(1-\gamma)}{\alpha} \mathbb{E}_{d_0(s),\pi(a|s)}[Q(s,a)]$$

$$+ \mathbb{E}_{s,a \sim d}\left[ (r(s,a) + \gamma \sum_{s'} p(s'|s,a)\pi(a'|s')Q(s',a') - Q(s,a))/\alpha \right] - D_{\mathrm{f}}(d(s,a) \,||\, d^O(s,a)) \tag{27}$$

$$= \max_{\pi(a|s)} \min_{Q(s,a)} \frac{(1-\gamma)}{\alpha} \mathbb{E}_{d_0(s),\pi(a|s)}[Q(s,a)]$$

$$+ \mathbb{E}_{s,a \sim d^O}\left[ f^*((r(s,a) + \gamma \sum_{s'} p(s'|s,a)\pi(a'|s')Q(s',a') - Q(s,a))/\alpha) \right] \tag{28}$$

The last step is due to the application of Eq 16 (convex conjugate definition). To see this more clearly let $y(s,a) = r(s,a) + \gamma \sum_{s'} p(s'|s,a)\pi(a'|s')Q(s',a') - Q(s,a)$. Then,

$$\max_{d \geq 0} \mathbb{E}_{s,a \sim d}[y(s,a)] - D_{\mathrm{f}}(d(s,a) \,||\, d^o(s,a)) \tag{29}$$

$$= \max_{d \geq 0} \mathbb{E}_{s,a \sim d^o}\left[ \frac{d(s,a)}{d^O(s,a)}y(s,a) - f\left(\frac{d(s,a)}{d^o(s,a)}\right) \right] \tag{30}$$

$$= \mathbb{E}_{d^o}[f^*(y(s,a))] \tag{31}$$

Finally, the policy optimization problem is reduced to the solving the following min-max optimization, which we will refer to as `dual-Q`:

$$\max_{\pi(a|s)} \min_{Q(s,a)} \frac{(1-\gamma)}{\alpha} \mathbb{E}_{d_0(s),\pi(a|s)}[Q(s,a)] + \mathbb{E}_{s,a \sim d^o}\left[ f^*((r(s,a) + \gamma \sum_{s'} p(s'|s,a)\pi(a|s')Q(s',a') - Q(s,a))/\alpha) \right] \tag{32}$$

For common $f$-divergences, table 2 lists the corresponding convex conjugates $f^*$. Also, note that the primal RL problem is convex and due to slater's condition [3] we can interchange the min-max between the Lagrange variable $Q$ and visitation distribution $d$ to max-min.

In the case of deterministic policy and deterministic dynamics, the above-obtained optimization takes a simpler form:

$$\min_{Q(s,a)} \max_{\pi(a|s)} \frac{(1-\gamma)}{\alpha} \mathbb{E}_{\rho_0(s)}[Q(s,\pi(s))] + \mathbb{E}_{s,a \sim d^o}[f^*((r(s,a) + \gamma Q(s',\pi(s')) - Q(s,a))/\alpha)] \tag{33}$$

Now, we have seen how we can transform a regularized RL problem into its `dual-Q` form which uses Lagrange variables in the form of state-action functions. Interestingly, we can go further to transform the regularized RL problem into Lagrange variables (V) that only depend on the state, and in doing so we also get rid of the min-max optimization in the `dual-Q`.

Consider the regularized RL problem again (Eq 21). This time, we formulate the visitation constraints to depend solely on states rather than state-action pairs. We consider $\alpha = 1$ for sake of exposition. Interested readers can derive the result for $\alpha \neq 1$ as in the `dual-Q` case above. Hence, we are solving the following constrained optimization problem:

$$\max_{d \geq 0} \mathbb{E}_{d(s,a)}[r(s,a)] - D_{\mathrm{f}}(d(s,a) \,||\, d^O(s,a)) \tag{34}$$

$$\text{s.t} \quad \sum_{a \in \mathcal{A}} d(s,a) = (1-\gamma)d_0(s) + \gamma \sum_{s',a'} d(s',a')p(s|s',a') \tag{35}$$

As before we construct the Lagrangian dual to this problem. Note that our constraints now solely depend on $s$.

$$\max_{d \geq 0} \min_{V(s)} \mathbb{E}_{s \sim d(s,a)}[r(s,a)] - D_{\mathrm{f}}(d(s,a) \,||\, d^o(s,a)) \tag{36}$$

$$+ \sum_s V(s) \left( (1-\gamma)d_0(s) + \gamma \sum_{s',a'} d(s',a')p(s|s',a') - d(s,a) \right) \tag{37}$$

Using similar algebraic manipulations we used to obtain `dual-Q`, we get the `dual-V` formulation for policy optimization:

$$\max_{d(s,a) \geq 0} \min_{V(s)} \mathbb{E}_{s,a \sim d(s,a)}[r(s,a)] - D_{\mathrm{f}}(d(s,a) \,||\, d^O(s,a))$$

$$+ \sum_s V(s) \left( (1-\gamma)d_0(s) + \gamma \sum_{s',a'} d(s',a')p(s|s',a') - d(s,a) \right) \tag{38}$$

$$= \min_{V(s)} \max_{d(s,a) \geq 0} (1-\gamma)\mathbb{E}_{d_0(s)}[V(s)]$$

$$+ \mathbb{E}_{s,a \sim d} \left[ r(s,a) + \gamma \sum_{s'} p(s'|s,a)V(s') - V(s) \right] - D_{\mathrm{f}}(d(s,a) \,||\, d^O(s,a)) \tag{39}$$

$$= \min_{V(s)} \max_{d(s,a) \geq 0} (1-\gamma)\mathbb{E}_{d_0(s)}[V(s)] \tag{40}$$

$$+ \mathbb{E}_{s,a \sim d} \left[ r(s,a) + \gamma \sum_{s'} p(s'|s,a)V(s') - V(s) \right] - D_{\mathrm{f}}(d(s,a) \,||\, d^O(s,a)) \tag{41}$$

$$= \min_{V(s)} (1-\gamma)\mathbb{E}_{d_0(s)}[V(s)] + \mathbb{E}_{s,a \sim d^O} \left[ f^*\left( r(s,a) + \gamma \sum_{s'} p(s'|s,a)V(s') - V(s) \right) \right] \tag{42}$$

In summary, we have two methods for policy optimization given by:

> `dual-Q`: $\max_\pi \min_Q (1-\gamma)\mathbb{E}_{d_0(s),\pi(a|s)}[Q(s,a)]$
> $\qquad\qquad + \mathbb{E}_{s,a \sim d^o}[f^*(r(s,a) + \gamma \sum_{s'} p(s'|s,a)\pi(a'|s')Q(s',a') - Q(s,a))]$

and,

> `dual-V`: $\min_{V(s)} (1-\gamma)\mathbb{E}_{d_0(s)}[V(s)] + \mathbb{E}_{s,a \sim d^o}[f^*(r(s,a) + \gamma \sum_{s'} p(s'|s,a)V(s') - V(s))]$

The above derivations for dual of RL CoP - `dual-Q` and `dual-V` brings out some important observations

- `dual-Q` and `dual-V` present off-policy policy optimization solutions for regularized RL problems which requires sampling transitions only from the distribution the policy state-action visitation is being regularized against.

- The above property allows us to solve not only RL problems but also imitation problems by setting the reward function to be zero everywhere and $d^o$ to be the expert dataset, and also offline RL problems where we want to maximize reward with the constraint that our state-action visitation should not deviate too much from the replay buffer ($d^o$ = replay-buffer).

- `dual-V` formulation presents a way to solve the RL problem using a single optimization rather than a min-max optimization of the Q-CoP or standard RL formulation. V-CoP implicitly subsumes greedy policy maximization.

### A.1.3 Recovering the optimal policy in V-CoP

In the above derivations for dual-Q and dual-V we leveraged the fact that the closed form solution for optimizing $d$ is known and it could be written in the form of Eq 29. The value of $d^*$ can found using KKT conditions on Eq 29:

$$\frac{d^*(s,a)}{d^o(s,a)} = \max\left(0, (f')^{-1}\left(\frac{y(s,a)}{\alpha}\right)\right) \tag{43}$$

Using this ratio there are two ways to recover the optimal policy:

**Method 1: Maximum likelihood on expert visitation distribution**

Policy learning can be treated as maximizing the likelihood of optimal actions:

$$\max \mathbb{E}_{s,a\sim d^*}[\pi_\theta(a|s)] \tag{44}$$

Using importance sampling we can rewrite the optimization above in a more tractable form:

$$\max_\theta \mathbb{E}_{s,a\sim d^o}[w^*(s,a)\pi_\theta(a|s)] \tag{45}$$

This way of policy learning is similar to weighted behavior cloning, but suffers from the issue that policy is not optimized at state-actions where the expert does not visit, i.e $w^*(s,a) = 0$

**Method 2: Reverse KL matching on offline data distribution (Information Projection)**

To allow the policy to be optimized at all that states in the offline dataset we consider an alternate objective:

$$\min_\theta D_{\mathrm{KL}}(d^o(s)\pi_\theta(a|s) \,||\, d^o(s)\pi^*(a|s)) \tag{46}$$

The objective can be written in a form suitable for optimization as follows:

$$\min_\theta D_{\mathrm{KL}}(d^o(s)\pi_\theta(a|s) \,||\, d^o(s)\pi^*(a|s)) = \min_\theta \mathbb{E}_{s\sim d^o(s),a\sim\pi_\theta}\left[\log\frac{\pi_\theta(a|s)}{\pi^*(a|s)}\right] \tag{47}$$

$$= \min_\theta \mathbb{E}_{s\sim d^o(s),a\sim\pi_\theta}\left[\log\frac{\pi_\theta(a|s)d^*(s)d^o(s)\pi^o(a|s)}{\pi^*(a|s)d^*(s)d^o(s)\pi^o(a|s)}\right] \tag{48}$$

$$= \min_\theta \mathbb{E}_{s\sim d^o(s),a\sim\pi_\theta}\left[\log\frac{\pi_\theta(a|s)}{\pi^o(a|s)} - \log(w^*(s,a)) + \log\frac{d^*(s)}{d^o(s)}\right] \tag{49}$$

$$= \min_\theta \mathbb{E}_{s\sim d^o(s),a\sim\pi_\theta}[\log(\pi_\theta(a|s)) - \log(\pi^o(a|s)) - \log(w^*(s,a))] \tag{50}$$

This method recovers the optimal policy at the states present in the dataset but requires learning another policy $\pi^o(a|s)$ which can be obtained by behavior cloning the replay buffer.

### A.2 Positivity constraints in Dual RL

We have ignored an important consideration in the derivation of dual-RL methods in Section A.1.2 – the constraint that the distribution $d$ we are optimizing for in Q-CoP and V-CoP must be positive. Although this does not affect derivation for `dual-Q` as it is overconstrained and the distribution is guaranteed to be unique, it is imperative we consider this constraint in the `dual-V` setting. We will now modify the derivation for `dual-V` to incorporate these constraints.

$$\max_{d\geq 0} \mathbb{E}_{d(s,a)}[r(s,a)] - D_f(d(s,a) \,||\, d^O(s,a)) \tag{51}$$

$$\text{s.t} \sum_{a\in\mathcal{A}} d(s,a) = (1-\gamma)d_0(s) + \gamma\sum_{s',a'} d(s',a')p(s|s',a') \tag{52}$$

We arrive at the following equation using the steps in Section A.1.2 (see Equation 39).

$$= \min_{V(s)} \max_{d(s,a) \geq 0} (1-\gamma)\mathbb{E}_{d_0(s)}[V(s)]$$

$$+ \mathbb{E}_{s,a \sim d}\left[ r(s,a) + \gamma \sum_{s'} p(s'|s,a)V(s') - V(s) \right] - D_{\mathrm{f}}(d(s,a) \,||\, d^O(s,a)) \tag{53}$$

$$= \min_{V(s)} \max_{d(s,a) \geq 0} (1-\gamma)\mathbb{E}_{d_0(s)}[V(s)]$$

$$+ \mathbb{E}_{s,a \sim d^O}\left[ \frac{d(s,a)}{d^O(s,a)}(r(s,a) + \gamma \sum_{s'} p(s'|s,a)V(s') - V(s)) \right] - \mathbb{E}_{s,a \sim d^O}\left[ f(\frac{d(s,a)}{d^O(s,a)}) \right] \tag{54}$$

Let $w(s,a) = \frac{d(s,a)}{d^O(s,a)}$ and $r(s,a) + \gamma \sum_{s'} p(s'|s,a)V(s') - V(s)$ be denoted by $y(s,a)$. We have,

$$\min_{V(s)} \max_{d(s,a) \geq 0} (1-\gamma)\mathbb{E}_{d_0(s)}[V(s)] + \mathbb{E}_{s,a \sim d^O}[w(s,a)(y(s,a))] - \mathbb{E}_{s,a \sim d^O}[f(w(s,a))] \tag{55}$$

We now direct the attention to the inner maximization and find a closed-form solution under the constraint that $d(s,a) \geq 0$.

$$\max_{d(s,a)} \max_{\lambda \geq =0} \mathbb{E}_{s,a \sim d^O}[w(s,a)(y(s,a))] - \mathbb{E}_{s,a \sim d^O}[f(w(s,a))] + \sum_{s,a} \lambda(s,a)w(s,a) \tag{56}$$

where $\lambda$ is the Lagrangian dual parameter than ensures the positivity constraint. Since strong duality holds, we can use the KKT constraints to find the solutions $w^*(s,a)$ and $\lambda^*(s,a)$.

**Primal feasibility**: $w^* \geq 0 \ \forall s,a$
**Dual feasibility**: $\lambda^* \geq 0 \ \forall s,a$
**Stationarity**: $d^O(s,a)(f'(w^*(s,a)) + y(s,a) + \lambda^*(s,a)) = 0 \ \forall s,a$
**Complementary Slackness**: $w^*(s,a)\lambda^*(s,a) = 0 \ \forall s,a$

Using stationarity we have the following:

$$f'(w^*(s,a)) = y(s,a) + \lambda^*(s,a) \ \forall s,a \tag{57}$$

Now using complementary slackness only two cases are possible $w^*(s,a) > 0$ or $\lambda^*(s,a) > 0$. Combining both cases we arrive at the following solution for this constrained optimization:

$$w^*(s,a) = \max\left(0, f'^{-1}(y(s,a))\right) \tag{58}$$

We refer to the resulting function after plugging the solution for $w^*$ back as $f_p^*$.

$$f_p^*(s,a) = w^*(s,a)(y(s,a)) - f(w^*(s,a)) \tag{59}$$

Note that we get the original conjugate $f^*$ back if we do not consider the positivity constraints. i.e

$$f^*(s,a) = f'^{-1}(y(s,a))(y(s,a)) - f(f'^{-1}(y(s,a))) \tag{60}$$

Finally, we have the following optimization to solve for dual-V when considering the positivity constraints:

> dual-V (with positivity constraints): $\min_{V(s)}(1-\gamma)\mathbb{E}_{d_0(s)}[V(s)] + \mathbb{E}_{s,a \sim d^O}\left[f_p^*(y(s,a))\right]$

### A.3 Dual Connections to Reinforcement Learning

Our first result shows that Conservative Q-learning [34], an offline RL method primarily understood to prevent overestimation by learning a lower bounded Q function is actually a dual-Q method. The lemma below formalizes the above statement:

**Lemma 5.** *Conservative Q-Learning (CQL) is the dual of Q-CoP with the generator function $f = (t-1)^2$ (Pearson $\chi^2$) and when the regularization distribution is the replay buffer ($d^O = d^R$).*

In other words, CQL eventually solves the regularized RL problem (Q-CoP) in its dual form where the regularization is a particular form of $f$-divergence. This unification indicates that its better performance compared to the family of behavior-regularized offline RL methods [42, 12, 53], which solve the Q-CoP using approximate dynamic programming is likely due to the choice of $f$-divergence and more amenable optimization afforded by the dual formulation. The dual-Q formulation has been previously studied for online RL by the name AlgaeDICE [41] but not evaluated in the context of offline RL. Lemma 5 also suggests that CQL is a special case of AlgaeDICE.

Leveraging the dual form of **V-CoP** converts the policy improvement problem from a min-max two-player game to a single optimization, thus potentially making the optimization easier to solve [40]. We also note that an additional step needs to be performed to recover policies in `dual-V` which requires solving a supervised learning problem (see Appendix A.1.3). We first show that Extreme Q-Learning (X-QL) [15], a method for both online and offline RL based on the principle of *implicit maximization* in the value function space using Gumbel regression, can be reduced to a `dual-V` problem with a semi-gradient update rule (i.e `stop-gradient` $(r(s,a) + \gamma \sum_{s'} p(s'|s,a)V(s'))$) when $f$ is set to be the reverse-KL regularization. Here, *implicit maximization* refers to finding the extreme values of a distribution using only samples from the distribution. This insight obtained through duality, allows us to propose a class of algorithms extending X-QL, by choosing different functions $f$ which we show below to result in a family of implicit maximizers.

**Lemma 6.** *Extreme Q-Learning (X-QL) is the dual of V-CoP with $f$-divergence set to be the reverse Kullback-Liebler divergence with a semi-gradient update rule.*

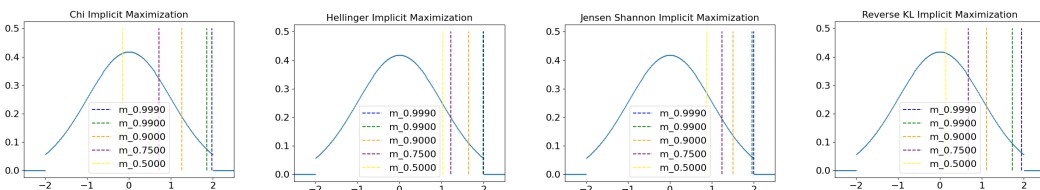

Figure 2: A family of implicit maximizers arising from semi-gradient dual reinforcement learning corresponding to different f-divergences. 10000 datapoints are sampled from 1-D bounded gaussian distribution $D$ and $v$ is inferred using Equation 62. As $\tau \to 1$ (see legend) we obtain more accurate estimates for the supremum of the support.

**A family of implicit maximizers**: Consider the $\lambda$-parameterized semi-gradient `dual-V` objective below:

$$\min_{V(s)}(1 - \lambda)\mathbb{E}_{d_O(s)}[V(s)] + \lambda\mathbb{E}_{s,a\sim d^O}\left[f_p^*\left(\left[\hat{Q}(s,a) - V(s))\right],\right)\right] \tag{61}$$

where $\hat{Q}(s,a) = r(s,a) + \gamma \sum_{s'} p(s'|s,a)V(s')$ with hat denoting stop-gradient. More generally for any random variable $X$ with distribution $D$,

$$\min_{v}(1 - \lambda)\mathbb{E}_{x\sim D}[v] + \lambda\mathbb{E}_{x\sim D}\left[f_p^*(x - v)\right]. \tag{62}$$

We show through Lemma 7 below and a simple example 2 that the semi-gradient form of `dual-V` optimization naturally gives rise to a family of implicit maximizers. Intuitively, this is because the second term in Eq 62 is minimized in value as $v$ increases and saturates once $v = \max(x \in support(D))$ while the first term is minimized for smaller $v$. This is opposed to specially curated implicit maximizers found in offline RL methods [32]. Gumbel regression becomes a special case of this family. We list some of the loss functions for value function updates with different $f$-divergences in Appendix A.3.1. We also highlight that the full-gradient variant of the `dual-V` framework for offline RL has been studied extensively in OptiDICE [35].

**Lemma 7.** *Let $X$ be a real-valued random variable with bounded support and the supremum of the support is $x^*$. Then optimizing equation 62, the solution $v_\lambda$ satisfies the following properties*

$$\lim_{\lambda\to 1} v_\lambda = x^* \text{ and } \forall \lambda_1 < \lambda_2 \in (0,1), \ v_{\lambda_1} \leq v_{\lambda_2}. \tag{63}$$

**A generalized policy iteration view of semi-gradient `dual-V`:** `dual-V` framework presents optimization difficulties when using full-gradients [4]. X-QL shows that stable learning can be achieved using the semi-gradient form. Our insight into implicit maximizers suggests that using semi-gradients brings the `dual-V` framework closer to generalized policy-iteration framework. The

update of $V$ in its semi-gradient dual form acts as an implicit policy optimizer and the estimation of $\hat{Q}(s, a)$ by regressing to $r(s, a) + \gamma(V(s'))$ is akin to a policy evaluation step, bridging the connection to generalized policy iteration.

**Proofs for this section:**

**Lemma 5.** *Conservative Q-Learning (CQL) is the dual of Q-CoP with the generator function $f = (t - 1)^2$ (Pearson $\chi^2$) and when the regularization distribution is the replay buffer ($d^O = d^R$).*

*Proof.* We show that CQL [34], a popular offline RL method is also a special case of `dual-Q` for offline RL. Consider an $f$-divergence with the generator function $f = (t - 1)^2$. The dual function $f^*$ is given by $f^* = (\frac{t^2}{4} + t)$. With the $f$-divergence our Q-CoP can be written as:

$$\frac{(1 - \gamma)}{\alpha} \mathbb{E}_{d_0, \pi(a|s)}[Q(s, a)] + \mathbb{E}_{s,a \sim d^O}\left[\frac{y(s, a, r, s')^2}{4\alpha^2} + \frac{y(s, a, r, s')}{\alpha}\right] \tag{64}$$

$$= \frac{(1 - \gamma)}{\alpha} \mathbb{E}_{d_0, \pi(a|s)}[Q(s, a)] + \mathbb{E}_{s,a \sim d^O}\left[\frac{y(s, a, r, s')}{\alpha}\right] + \mathbb{E}_{s,a \sim d^O}\left[\frac{y(s, a, r, s')^2}{4\alpha^2}\right] \tag{65}$$

Let's simplify the first two terms:

$$\frac{1}{\alpha}\left[(1 - \gamma)\mathbb{E}_{d_0, \pi(a|s)}[Q(s, a)] + \mathbb{E}_{s,a \sim d^O}\left[r(s, a) + \gamma \sum_{s',a'} p(s'|s, a)\pi(a'|s')Q(s', a') - Q(s, a)\right]\right] \tag{66}$$

$$= \frac{1}{\alpha}\left[(1 - \gamma)\mathbb{E}_{d_0, \pi(a|s)}[Q(s, a)] + \mathbb{E}_{s,a \sim d^O}\left[\gamma \sum_{s',a'} p(s'|s, a)\pi(a'|s')Q(s', a')\right] - \mathbb{E}_{s,a \sim d^O}[Q(s, a)] + \cancel{\mathbb{E}_{s,a \sim d^O}[r(s, a)]}\right] \tag{67}$$

$$= \frac{1}{\alpha}\left[(1 - \gamma)\sum_{s,a} d_0(s)\pi(a|s)Q(s, a) + \gamma \sum_{s,a} d^O(s, a)\sum_{s'} p(s'|s, a)\pi(a'|s')Q(s', a') - \mathbb{E}_{s,a \sim d^O}[Q(s, a)]\right] \tag{68}$$

$$= \frac{1}{\alpha}\left[(1 - \gamma)\sum_{s,a} d_0(s)\pi(a|s)Q(s, a) + \gamma\langle d^O, P^\pi Q\rangle - \mathbb{E}_{s,a \sim d^O}[Q(s, a)]\right] \tag{69}$$

$$= \frac{1}{\alpha}\left[(1 - \gamma)\sum_{s,a} d_0(s)\pi(a|s)Q(s, a) + \gamma\langle P_*^\pi d^o, Q\rangle - \mathbb{E}_{s,a \sim d^o}[Q(s, a)]\right] \tag{70}$$

$$= \frac{1}{\alpha}\left[(1 - \gamma)\sum_{s,a} d_0(s)\pi(a|s)Q(s, a) + \gamma \sum_{s,a} \pi(a|s)Q(s, a)\sum_{s',a'} p(s|s', a')d(s', a') - \mathbb{E}_{s,a \sim d^o}[Q(s, a)]\right] \tag{71}$$

$$= \frac{1}{\alpha}\left[\sum_{s,a}(d_0(s) + \gamma \sum_{s'a,'} p(s|s', a')d(s', a'))\pi(a|s)Q(s, a) - \mathbb{E}_{s,a \sim d^o}[Q(s, a)] + \mathbb{E}_{s,a \sim d^o}[r(s, a)]\right] \tag{72}$$

$$= \frac{1}{\alpha}\left[\sum_{s,a} d^o(s)\pi(a|s)Q(s, a) - \mathbb{E}_{s,a \sim d^o}[Q(s, a)] + \mathbb{E}_{s,a \sim d^o}[r(s, a)]\right] \tag{73}$$

$$= \frac{1}{\alpha}\left[\mathbb{E}_{s \sim d^o, a \sim \pi}[Q(s, a)] - \mathbb{E}_{s,a \sim d^o}[Q(s, a)]\right] \tag{74}$$

where $P^\pi$ denotes the policy transition operator, $P^\pi_*$ denotes the adjoint policy transition operator. Removing constant terms (Equation 67) with respect to optimization variables we end up with the following form for `dual-Q`:

$$\frac{1}{\alpha}\left[\mathbb{E}_{s\sim d^o,a\sim\pi}[Q(s,a)] - \mathbb{E}_{s,a\sim d^o}[Q(s,a)]\right] + \mathbb{E}_{s,a\sim d^o}\left[\frac{y(s,a,r,s')^2}{4\alpha^2}\right] \tag{75}$$

Hence the `dual-Q` optimization reduces to:

$$\max_\pi \min_Q \alpha\left[\mathbb{E}_{s\sim d^o,a\sim\pi}[Q(s,a)] - \mathbb{E}_{s,a\sim d^o}[Q(s,a)]\right] + \mathbb{E}_{s,a\sim d^o}\left[\frac{y(s,a,r,s')^2}{4}\right] \tag{76}$$

This equation matches the unregularized CQL objective (Equation 3 in [34]). $\qquad\square$

**Lemma 6.** *Extreme Q-Learning (X-QL) is the dual of V-CoP with $f$-divergence set to be the reverse Kullback-Liebler divergence with a semi-gradient update rule.*

*Proof.* We show that the Extreme Q-Learning [] framework is a special case of the dual framework, specifically the `dual-V` using the semi-gradient update rule.

Consider setting the $f$-divergence to be the KL divergence in the dual V framework, the regularization distribution and the initial state distribution to be the replay buffer distribution ($d^O = d^R$ and $d_0 = d^R$). The conjugate of the generating function for KL divergence is given by $f^*(t) = e^{t-1}$.

$$\min_{V(s)}(1-\gamma)\mathbb{E}_{d_0(s)}[V(s)] + \mathbb{E}_{s,a\sim d^R}\left[f^*\left(\left[r(s,a)+\gamma\sum_{s'}p(s'|s,a)V(s')-V(s))\right]/\alpha\right)\right] \tag{77}$$

$$\min_{V(s)}(1-\gamma)\mathbb{E}_{d_0(s)}[V(s)] + \mathbb{E}_{s,a\sim d^R}\left[\exp(\left(\left[r(s,a)+\gamma\sum_{s'}p(s'|s,a)V(s')-V(s))\right]/\alpha-1\right)\right] \tag{78}$$

A popular approach for stable optimization in temporal difference learning is the semi-gradient update rule which has been studied in previous works []. In this update strategy, we fix the targets for the temporal difference backup. Our target in the above optimization is given by:

$$\hat{Q}(s,a) = r(s,a) + \gamma\sum_{s'}p(s'|s,a)V(s') \tag{79}$$

The update equation for V is now given by:

$$\min_{V(s)}(1-\gamma)\mathbb{E}_{d_0(s)}[V(s)] + \mathbb{E}_{s,a\sim d^R}\left[\exp(\left(\left[\hat{Q}(s,a)-V(s))\right]/\alpha-1\right)\right] \tag{80}$$

where hat denotes the `stop-gradient` operation. We approximate this target by using mean-squared regression with the single sample unbiased estimate as follows:

$$\min_Q \mathbb{E}_{s,a,s'\sim d^R}\left[(Q(s,a)-(r(s,a)+V(s')))^2\right] \tag{81}$$

The procedure is now equivalent to Extreme-Q learning and is a special case of the `dual-V` framework.

$\qquad\square$

### A.3.1  A family of implicit maximizers

**Lemma 7.** *Let $X$ be a real-valued random variable with bounded support and the supremum of the support is $x^*$. Then optimizing equation 62, the solution $v_\lambda$ satisfies the following properties*

$$\lim_{\lambda \to 1} v_\lambda = x^* \text{ and } \forall \, \lambda_1 < \lambda_2 \in (0,1), \ v_{\lambda_1} \le v_{\lambda_2}. \tag{63}$$

*Proof.* We analyze the behavior of the following optimization of interest.

$$\min_v (1 - \lambda) \mathbb{E}_{x \sim D}[v] + \lambda \mathbb{E}_{x \sim D}\left[f_p^*(x - v)\right] \tag{82}$$

$f_p^*(t)$ is given by (using the result derived in A.2):

$$f_p^*(t) = -f\left(\max(f'^{-1}(t), 0)\right) + t \max\left(f'^{-1}(t), 0\right) \tag{83}$$

The function $f_p^*$ admits two different behaviors given by:

$$f_p^* = \begin{cases} -f(f'^{-1}(t)) + t f'^{-1}(t) = f^*(t), & \text{if } f'^{-1}(t) > 0 \\ -f(0), & \text{otherwise} \end{cases}$$

where $f^*$ is the convex conjugate of $f$-divergence and is strictly increasing with $t$. We note other properties related to $f$ function for $f$-divergences: $f^*, f', (f')^{-1}$ is strictly increasing, $f(0_+) > 0$ and $(f')^{-1}(t) > 0$ when $t > 0$ and $0$ otherwise.

We analyze the second term in Eq 82. It can be expanded as follows:

$$\lambda \int_{x:(f')^{-1}(x-v)>0} p(x) f^*(x-v) dx - \lambda \int_{x:(f')^{-1}(x-v)<0} f(0) p(x) dx \tag{84}$$

From the properties of $f$, we use the fact that $(f')^{-1}(x - v) > 0$ when $x - v > 0$ or equivalently $x > v$.

$$\lambda \int_{x>v} p(x) f^*(x-v) dx - \lambda \int_{x \le v} f(0) p(x) dx \tag{85}$$

The first term in the above equation decreases monotonically and the second term increases monotonically (thus the combined terms decrease) as $v$ increases until $v = x^*$ (supremum of the support of the distribution) after which the equation assumes a constant value of $-\lambda f(0)$.

Going back to our original optimization in Equation 82, the first term decreases monotonically with $v$. As $\lambda \to 1$, the minimization of the second term takes precedence, with increasing v until saturation ($v = x^*$). We can go further to characterize the effect of $\lambda$ on solution $v_\lambda$ of the equation. The solution of the optimization can be written in closed form as:

$$\frac{(1 - \lambda)}{\lambda} = \mathbb{E}_{x \sim D}\left[f_p^{*'}(x - v)\right] \tag{86}$$

Using the fact that $f_p^{*'}$ is non-decreasing, we can show that the right-hand term in the equation above increases as $v$ decreases. This in turn implies that for all $\lambda_1, \lambda_2$ such that $\lambda_1 \le \lambda_2$ we have that $v_{\lambda_1} \le v_{\lambda_2}$ .

$\square$

### A.4  Dual Connections to Imitation Learning

### A.4.1  Offline imitation learning with expert data only

**A new method for offline imitation learning:**  Analogous to `dual-Q` (offline imitation), we can leverage the `dual-V` (offline imitation) setting which avoids the min-max optimization given by:

`IV-Learn` or `dual-V` (offline imitation from expert-only data):

$$\min_{V(s)} (1 - \gamma) \mathbb{E}_{d_0(s)}[V(s)] + \mathbb{E}_{s,a \sim d^E}[f^*\left(\left[\mathcal{T}_0 V(s,a) - V(s)\right]/\alpha\right)] \tag{87}$$

We propose `dual-V` (offline imitation) to be a new method arising out of this framework which we leave for future exploration.

**Proofs for this section:**

**Corollary 1.** `dual-Q` *is equivalent to Implicit Behavior Cloning [7] when $r(s,a) = 0$ $\forall$ $(\mathcal{S}, \mathcal{A})$ and $d^O(s,a) = d^E(s,a)$ and $f$ is set to be the total variation divergence.*

Equation 10 suggests that intuitively IQ-learn trains an energy-based model in the form of Q where it pushes down the Q-values for actions predicted by current policy and pushes up the Q-values at the expert state-action pairs. This becomes more clear when the divergence $f$ is chosen to be Total-Variation ($f^* = \mathbb{I}$), IQ learn reduces to:

$$(1-\gamma)\mathbb{E}_{d_0(s),\pi(a|s)}[Q(s,a)] + \mathbb{E}_{s,a\sim d^E}\left[\gamma\sum_{s',a'} p(s'|s,a)\pi(a'|s')Q(s',a') - Q(s,a)\right] \tag{88}$$

$$= \left[(1-\gamma)\mathbb{E}_{d_0(s),\pi(a|s)}[Q(s,a)] + \mathbb{E}_{s,a\sim d^E}\left[\gamma\sum_{s',a'} p(s'|s,a)\pi(a'|s')Q(s',a')\right]\right] - \mathbb{E}_{s,a\sim d^E}[Q(s,a)] \tag{89}$$

Let's simplify the first two terms:

$$(1-\gamma)\mathbb{E}_{d_0(s),\pi(a|s)}[Q(s,a)] + \mathbb{E}_{s,a\sim d^E}\left[\gamma\sum_{s'} p(s'|s,a)\pi(a'|s')Q(s',a')\right] \tag{90}$$

$$= (1-\gamma)\sum_{s,a} d_0(s)\pi(a|s)Q(s,a) + \gamma\sum_{s,a} d^E(s,a)\sum_{s',a'} p(s'|s,a)\pi(a'|s')Q(s',a') \tag{91}$$

$$= (1-\gamma)\sum_{s,a} d_0(s)\pi(a|s)Q(s,a) + \gamma\sum_{s',a'}\sum_{s,a} d^E(s,a)p(s'|s,a)\pi(a'|s')Q(s',a') \tag{92}$$

$$= (1-\gamma)\sum_{s,a} d_0(s)\pi(a|s)Q(s,a) + \gamma\sum_{s',a'} \pi(a'|s')Q(s',a')(\sum_{s,a} d^E(s,a)p(s'|s,a)) \tag{93}$$

$$= (1-\gamma)\sum_{s,a} d_0(s)\pi(a|s)Q(s,a) + \gamma\sum_{s',a'} \pi(a'|s')Q(s',a')(\sum_{s,a} d^E(s,a)p(s'|s,a)) \tag{94}$$

$$= (1-\gamma)\sum_{s,a} d_0(s)\pi(a|s)Q(s,a) + \gamma\sum_{s,a} \pi(a|s)Q(s,a)(\sum_{s',a'} d^E(s',a')p(s|s',a')) \tag{95}$$

$$= \sum_{s,a}(1-\gamma)d_0(s)\pi(a|s)Q(s,a) + \pi(a|s)Q(s,a)(\sum_{s',a'} d^E(s',a')p(s|s',a')) \tag{96}$$

$$= \sum_{s,a} \pi(a|s)Q(s,a)\left[(1-\gamma)d_0(s) + \gamma\sum_{s',a'} d^E(s',a')p(s|s',a')\right] \tag{97}$$

$$= \sum_{s,a} \pi(a|s)Q(s,a)d^E(s) \tag{98}$$

where the last step is due to the steady state property of the MDP (Bellman flow constraint).

Therefore IQ-Learn/`dual-Q` for offline imitation (in the special case of TV divergence) simplifies to (from Equation 89):

$$\left[(1-\gamma)\mathbb{E}_{d_0(s),\pi(a|s)}[Q(s,a)] + \mathbb{E}_{s,a\sim d^E}\left[\gamma\sum_{s',a'}p(s'|s,a)\pi(a'|s')Q(s',a')\right]\right] - \mathbb{E}_{s,a\sim d^E}[Q(s,a)]$$

(99)

$$= \min_Q \mathbb{E}_{d_E(s),\pi(a|s)}[Q(s,a)] - \mathbb{E}_{s,a\sim d^E}[Q(s,a)]$$

(100)

The gradient w.r.t for the above optimization matches the gradient update of Implicit Behavior Cloning [7] with $Q$ as the energy-based model.

## A.5 Off-policy imitation learning (under coverage assumption)

First, we show that is easy to see why choosing the $f$-divergence to be reverse KL makes it possible to get an off-policy objective for imitation learning in the dual framework. We start with the Q-CoP for imitation learning using the reverse KL-divergence ($r(s,a) = 0$ and $d^o = d^E$):

$$\max_{d(s,a)\geq 0,\pi(a|s)} -D_{\text{KL}}(d(s,a) \,||\, d^E(s,a))$$

$$\text{s.t } d(s,a) = (1-\gamma)\rho_0(s).\pi(a|s) + \gamma\pi(a|s)\sum_{s',a'}d(s',a')p(s|s',a').$$

(101)

*Under the assumption that the replay buffer visitation (denoted by $d^R$) covers the expert visitation ($d^R > 0$ wherever $d^E > 0$) [37], which we refer to as the* **coverage assumption**, *the reverse KL divergence can be expanded as follows:*

$$D_{\text{KL}}(d(s,a) \,||\, d^E(s,a)) = \mathbb{E}_{s,a\sim d(s,a)}\left[\log\frac{d(s,a)}{d^E(s,a)}\right] = \mathbb{E}_{s,a\sim d(s,a)}\left[\log\frac{d(s,a)}{d^E(s,a)}\frac{d^R(s,a)}{d^R(s,a)}\right]$$

(102)

$$= \mathbb{E}_{s,a\sim d(s,a)}\left[\log\frac{d(s,a)}{d^R(s,a)} + \log\frac{d^R(s,a)}{d^E(s,a)}\right]$$

(103)

$$= \mathbb{E}_{s,a\sim d(s,a)}\left[\log\frac{d^R(s,a)}{d^E(s,a)}\right] + D_{\text{KL}}(d(s,a) \,||\, d^R(s,a)).$$

(104)

Hence the Q-CoP can now be written as:

$$\max_{d(s,a)\geq 0,\pi(a|s)} \mathbb{E}_{s,a\sim d(s,a)}\left[-\log\frac{d^R(s,a)}{d^E(s,a)}\right] - D_{\text{KL}}(d(s,a) \,||\, d^R(s,a))$$

(105)

$$\text{s.t } d(s,a) = (1-\gamma)\rho_0(s).\pi(a|s) + \gamma\sum_{s',a'}d(s',a')p(s|s',a')\pi(a|s).$$

(106)

Now, in the optimization above the first term resembles the reward function and the second term resembles the divergence constraint with a new distribution $d^R(s,a)$ in the original regularized RL primal (Eq 22). Hence we can obtain respective `dual-Q` and `dual-V` in the setting for off-policy imitation learning using the reward function as $r^{imit}(s,a) = -\log\frac{d^R(s,a)}{d^E(s,a)}$ and the new regularization distribution as $d^R(s,a)$. Using $\mathcal{T}^{\pi}_{r^{imit}}$ and $\mathcal{T}_{r^{imit}}$ to denote backup operators under new reward function $r^{imit}$, we have

`dual-Q` for off-policy imitation (coverage assumption):

$$\max_{\pi(a|s)} \min_{Q(s,a)} (1-\gamma)\mathbb{E}_{\rho_0(s),\pi(a|s)}[Q(s,a)] + \mathbb{E}_{s,a\sim d^R}[f^*(\mathcal{T}^{\pi}_{r^{imit}}Q(s,a) - Q(s,a))].$$

(107)

This choice of KL divergence leads us to a reduction of another method, OPOLO [57] for off-policy imitation learning to `dualQ` which we formalize in the lemma below:

**Lemma 8.** `dual-Q` *for off-policy imitation learning reduces to OPOLO [57], with the $f$-divergence set to the reverse KL divergence when $r(s,a) = 0 \,\forall\mathcal{S},\mathcal{A}, d^O = d^E$ and under the assumption that the replay data distribution covers the expert data distribution.*

Analogously we have `dual-V` for off-policy imitation (coverage assumption):

$$\min_{V(s)}(1-\gamma)\mathbb{E}_{\rho_0(s)}[V(s)] + \mathbb{E}_{s,a\sim d^R}[f^*(\mathcal{T}_{r^{imit}}V(s,a) - V(s))]. \tag{108}$$

We note that the `dual-V` framework for off-policy imitation learning under coverage assumptions was studied in the imitation learning work SMODICE [37].

## B    Off-policy imitation learning with relaxed coverage

We now derive our proposed method for imitation learning with arbitrary data. The derivation for the `dual-Q` setting is shown below. `dual-V` derivation can be done similarly.

**Lemma 3.** (***dual-Q** for off-policy imitation (relaxed coverage assumption)*) *Imitation learning using off-policy data can be solved by optimizing the following modified dual objective for Q-CoP with $r(s,a) = 0 \,\forall \mathcal{S}, \mathcal{A}$ and $f$-divergence considered between distributions $d^R_{mix} := \beta d(s,a) + (1 - \beta)d^R(s,a)$ and $d^{E,R}_{mix} := \beta d^E(s,a) + (1 - \beta)d^R(s,a)$, and is given by:*

$$\max_{\pi(a|s)} \min_{Q(s,a)} \beta(1-\gamma)\mathbb{E}_{d_0(s),\pi(a|s)}[Q(s,a)] + \mathbb{E}_{s,a\sim d^{E,R}_{mix}}\left[f_p^*(\mathcal{T}_0^\pi Q(s,a) - Q(s,a))\right]$$
$$- (1-\beta)\mathbb{E}_{s,a\sim d^R}[\mathcal{T}_0^\pi Q(s,a) - Q(s,a)] \tag{14}$$

*Proof.*

$$\max_{\pi,d\geq 0} \min_{Q(s,a)} \alpha\mathbb{E}_{s,a\sim d}[r(s,a)] - D_f(d^R_{mix} \,||\, d^{E,R}_{mix})$$
$$+ \alpha\sum_{s,a} Q(s,a)\left((1-\gamma)d_0(s).\pi(a|s) + \gamma\sum_{s',a'} d(s',a')p(s|s',a')\pi(a|s) - d(s,a)\right)$$

We can use the same algebraic machinery as before (Section A.1.2) to get an unconstrained tractable optimization problem:

$$\max_{\pi,d\geq 0} \min_{Q(s,a)} \alpha\mathbb{E}_{s,a\sim d(s,a)}[r(s,a)] - D_f(d^R_{mix} \,||\, d^{E,R}_{mix})$$
$$+ \alpha\sum_{s,a} Q(s,a)\left((1-\gamma)d_0(s).\pi(a|s) + \gamma\sum_{s',a'} d(s',a')p(s|s',a')\pi(a|s) - d(s,a)\right) \tag{109}$$

$$= \max_{\pi,d\geq 0} \min_{Q(s,a)} \alpha(1-\gamma)\mathbb{E}_{d_0(s),\pi(a|s)}[Q(s,a)]$$
$$+ \alpha\mathbb{E}_{s,a\sim d}\left[r(s,a) + \gamma\sum_{s'} p(s'|s,a)\pi(a'|s')Q(s',a') - Q(s,a)\right] - D_f(d^R_{mix} \,||\, d^{E,R}_{mix}) \tag{110}$$

$$= \max_{\pi,d\geq 0} \min_{Q(s,a)} \alpha(1-\gamma)\mathbb{E}_{d_0(s),\pi(a|s)}[Q(s,a)]$$
$$+ \alpha\mathbb{E}_{s,a\sim d}\left[r(s,a) + \gamma\sum_{s'} p(s'|s,a)\pi(a'|s')Q(s',a') - Q(s,a)\right]$$
$$+ (1-\alpha)\mathbb{E}_{s,a\sim d^R}\left[r(s,a) + \gamma\sum_{s'} p(s'|s,a)\pi(a'|s')Q(s',a') - Q(s,a)\right]$$
$$- (1-\alpha)\mathbb{E}_{s,a\sim d^R}\left[r(s,a) + \gamma\sum_{s'} p(s'|s,a)\pi(a'|s')Q(s',a') - Q(s,a)\right] - D_f(d^R_{mix} \,||\, d^{E,R}_{mix})$$
$$\tag{111}$$

Note that the inner maximization with respect to $d$ has the constraint that $d \geq 0$. This constraint was not necessary for the previous settings for `dual-Q` problems we have discussed. In this setting, to get a tractable closed form we replace the optimization variable from $d$ to $d_{mix}^R$ with the constraint that $d \geq 0$. This prevents the optimization to result in values for $d_{mix}^R$ which has $d < 0$. Ignoring this constraint ($d \geq 0$) results in the following dual-optimization for imitation from arbitrary data.

$$\max_{\pi(a|s)} \min_{Q(s,a)} \alpha(1-\gamma)\mathbb{E}_{d_0(s),\pi(a|s)}[Q(s,a)]$$

$$+ \mathbb{E}_{s,a \sim d_{mix}^{E,R}}\left[f^*\left(r(s,a) + \gamma \sum_{s'} p(s'|s,a)\pi(a'|s')Q(s',a') - Q(s,a)\right)\right]$$

$$- (1-\alpha)\mathbb{E}_{s,a \sim d^R}\left[r(s,a) + \gamma \sum_{s'} p(s'|s,a)\pi(a'|s')Q(s',a') - Q(s,a)\right] \qquad (113)$$

To incorporate the positivity constraints we begin on the inner maximization w.r.t $d_{mix}^R$ and consider the terms dependent on $d_{mix}^R$ below.

$$\max_{d_{mix}^R, d \geq 0} \mathbb{E}_{s,a \sim d_{mix}^R}\left[r(s,a) + \gamma \sum_{s'} p(s'|s,a)\pi(a'|s')Q(s',a') - Q(s,a)\right] - D_{\mathrm{f}}(d_{mix}^R \,||\, d_{mix}^{E,R})$$

$$(114)$$

Let $p(s,a) = \frac{(1-\alpha)\rho^R(s,a)}{\alpha\rho^E(s,a)+(1-\alpha)\rho^R(s,a)}$, $y(s,a) = r(s,a)+\gamma \sum_{s'} p(s'|s,a)\pi(a'|s')Q(s',a')-Q(s,a)$ and $w(s,a) = \frac{d_{mix}^R(s,a)}{d_{mix}^E(s,a)}$. We construct the lagrangian dual to incorporate the constraint $d \geq 0$ in its equivalent form $w(s,a) \geq p(s,a)$ and obtain the following:

$$\max_{w(s,a)} \max_{\lambda \geq 0} \mathbb{E}_{s,a \sim d_{mix}^{E,R}}[w(s,a)y(s,a)] - \mathbb{E}_{d_{mix}^{E,R}}[f(w(s,a))] + \sum_{s,a} \lambda(w(s,a) - p(s,a)) \qquad (115)$$

Since strong duality holds, we can use the KKT constraints to find the solutions $w^*(s,a)$ and $\lambda^*(s,a)$.

**Primal feasibility**: $w^*(s,a) \geq p(s,a) \ \forall\, s,a$
**Dual feasibility**: $\lambda^* \geq 0 \ \forall\, s,a$
**Stationarity**: $d_{mix}^{E,R}(s,a)(f'(w^*(s,a)) + y(s,a) + \lambda^*(s,a)) = 0 \ \forall\, s,a$
**Complementary Slackness**: $(w^*(s,a) - p(s,a))\lambda^*(s,a) = 0 \ \forall\, s,a$

Using stationarity we have the following:

$$f'(w^*(s,a)) = y(s,a) + \lambda^*(s,a) \ \forall\, s,a \qquad (116)$$

Now using complementary slackness only two cases are possible $w^*(s,a) > p(s,a)$ or $\lambda^*(s,a) > 0$. Combining both cases we arrive at the following solution for this constrained optimization:

$$w^*(s,a) = \max\left(p(s,a), f'^{-1}(y(s,a))\right) \qquad (117)$$

We can still find a closed-form solution for the inner optimization, in the case when $d \geq 0$, although a bit more involved (See Appendix for the proof). Let $y(s, a) = r(s, a) + \gamma \sum_{s'} p(s'|s, a)\pi(a'|s')Q(s', a') - Q(s, a)$. Also let $p(s, a) = \frac{(1-\alpha)\rho^R(s,a)}{\alpha\rho^E(s,a)+(1-\alpha)\rho^R(s,a)}$.

$$
\max_{\pi(a|s)} \min_{Q(s,a)} \alpha(1-\gamma)\mathbb{E}_{d_0(s),\pi(a|s)}[Q(s, a)]
$$
$$
+ \mathbb{E}_{s,a \sim d_{mix}^{E,R}}\left[\max\left(p(s,a), (f')^{-1}\left(y(s,a)\right)\right)y(s,a) - \alpha f\left(\max\left(p(s,a), (f')^{-1}\left(y(s,a)\right)\right)\right)\right]
$$
$$
\tag{118}
$$

$$
- (1-\alpha)\mathbb{E}_{s,a \sim d^R}\left[r(s,a) + \gamma \sum_{s'} p(s'|s,a)\pi(a'|s')Q(s',a') - Q(s,a)\right] \tag{119}
$$

Thus, the closed-form solution with the positivity constraints requires us to estimate the ratio $p(s, a)$ which is possible by learning a discriminator. We observed in our experiments that ignoring the positivity constraints still resulted in a performant method while having the benefits of being simple. A similar derivation can be done in V-space to obtain an analogous result. $\qquad\square$

## C  Implementation and Experiment Details

**Environments:** In this work for benchmarking we use 4 MuJoCo (licensed under CC BY 4.0) locomotion environments: Hopper, Walker2d, HalfCheetah, and Ant. .

**Offline datasets**: In this task, we use offline dataset of environments interactions from D4RL [8]. We consider the following MuJoCo environments: Our dataset composition for 'random+expert' is similar to SMODICE [37] where we use a mixture of a small number of expert trajectories ($\leq 200$ trajectories) and a large number of low-quality trajectories from the "random-v2" dataset (1 million transitions). We similarly create another offline dataset 'medium+expert' consisting of 200 expert trajectories and 1 million medium-quality transitions from the "medium-v2". The 'random+few-expert' dataset is similar to the 'random+expert' dataset except that only 30 expert trajectories are present in the offline dataset.

**Expert dataset** The offline dataset for imitation consists of 1000 transitions obtained from the "expert-v2" dataset for the respective environment.

**Baselines:** We compare our proposed methods against 4 representative methods for offline imitation learning with suboptimal data – SMODICE [37], RCE [6], ORIL [59] and IQLearn [14]. We do not compare to DEMODICE [29] as SMODICE was shown to be competitive in [37]. SMODICE is an imitation method emerging from the dual framework but under an restrictive coverage assumption. ORIL adapts GAIL [21] to the offline setting by using an offline RL algorithm for policy optimization. RCE baseline in the paper combine RCE (Eysenbach et al., 2021), the state-of-art online example-based RL method, and TD3-BC. ORIL and RCE share the same state-action based discriminator as in SMODICE, and TD3-BC [11] as the offline RL algorithm. All the approaches only have access to expert state-action trajectory.

We use the author's open-source implementations of baselines SMODICE, RCE, ORIL available at `https://github.com/JasonMa2016/SMODICE`. We use the author-provided hyperparameters (similar to those used in [37]) for all MuJoCo locomotion environments. IQ-Learn was tested on our expert dataset by following authors implementation found here: `https://github.com/Div99/IQ-Learn`. We tested two IQ-Learn loss variants: 'v0' and 'value' as found in their hyperparameter configurations and took the best out of the two runs.

**Policy Optimization:** We use Method 1 in Section A.1.3 for policy update.

### C.1  Hyperparameters

Hyperparameters for our proposed offpolicy imitation learning method `ReCOIL` are shown in Table 3.

| Hyperparameter | Value |
|---|---|
| Policy updates $n_{pol}$ | 1 |
| Policy learning rate | 3e-5 |
| Value learning rate | 3e-4 |
| Temperature $\alpha$ | 0.1 |
| $f$-divergence | $\chi^2$ |

Table 3: Hyperparameters for `ReCOIL`.

# D Experimental Results

## D.1 The failure of ADP-based traditional off-policy algorithms

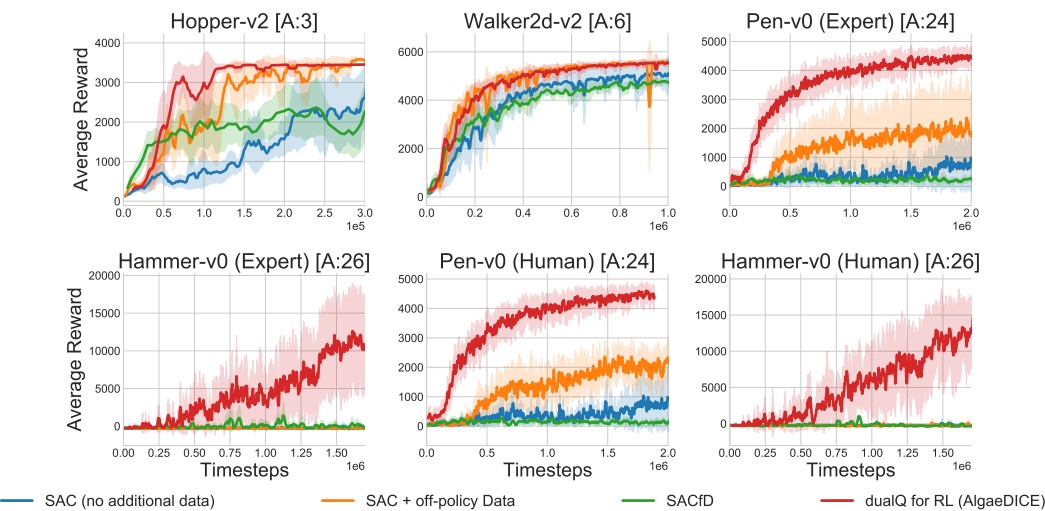

Figure 3: Despite the promise of off-policy methods, current methods based on ADP such as SAC fail when the dimension of action space, denoted by A, increases even when helpful data is added to their replay buffer. On other hand, dual-Q methods are able to leverage off-policy data to increase their learning performance

Figure 3 shows that methods like SAC, SACfD deteriorate increasingly with increasing action dimension when bootstrapped with off-policy data. Figure 4 shows that traditional ADP methods suffer from overestimation during training.

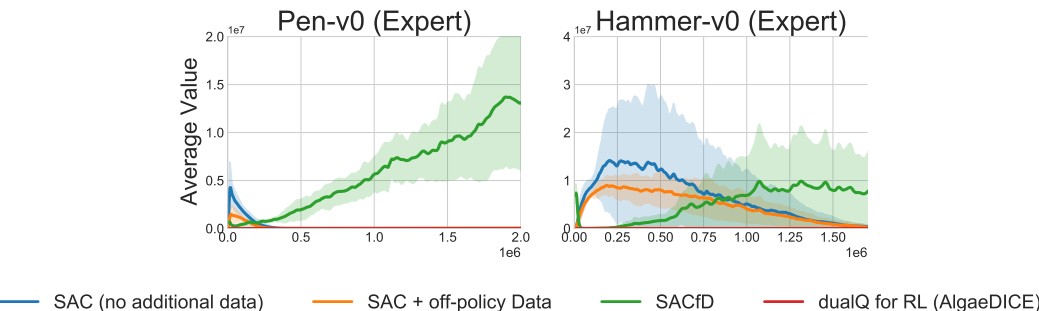

Figure 4: SAC and SACfD suffer from overestimation when off-policy data is added to the replay buffer. We hypothesize this to cause instabilities during training while dualQ has no overestimation.

## D.2 Does `ReCOIL` allow for better estimation of agent visitation distribution?

We consider two didactic environments which demonstrate the failures of method that either do not utilize off-policy data (IQ-Learn) or relies on a coverage assumption (SMODICE). ReCOIL is able to

perfectly infer agent's visitation when replay buffer covers agent ground truth visitation perfectly (Fig 5) and is able to outperform baselines when the replay buffer has imperfect coverage over the agent's ground truth visitation (Fig 6).

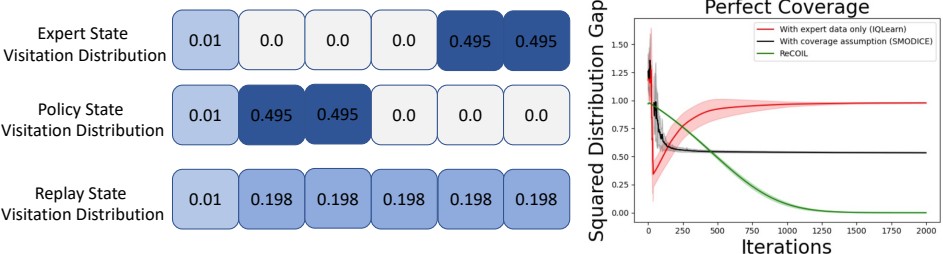

Figure 5: The replay buffer distribution covers the agent policy visitation distribution. Using `ReCOIL`, we are perfectly able to infer the agent policy visitation whereas a method that only relies on expert data or the replay data with the coverage assumption fails. Results are averaged over 100 seeds.

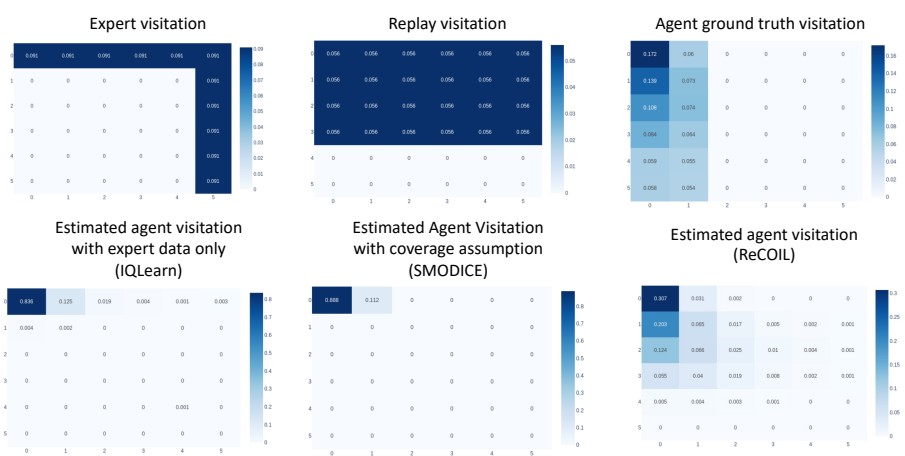

Figure 6: Replay buffer consists of data that visits near the initial state (0,0), a setting commonly observed when training RL agents. We estimate agent's policy visitation and observe `ReCOIL` to outperform both methods which rely on expert data only or use the replay data with coverage assumption

### D.3   Benchmarking performance of ReCOIL on MuJoCo tasks

We show learning curves for ReCOIL in Figure 7 below.

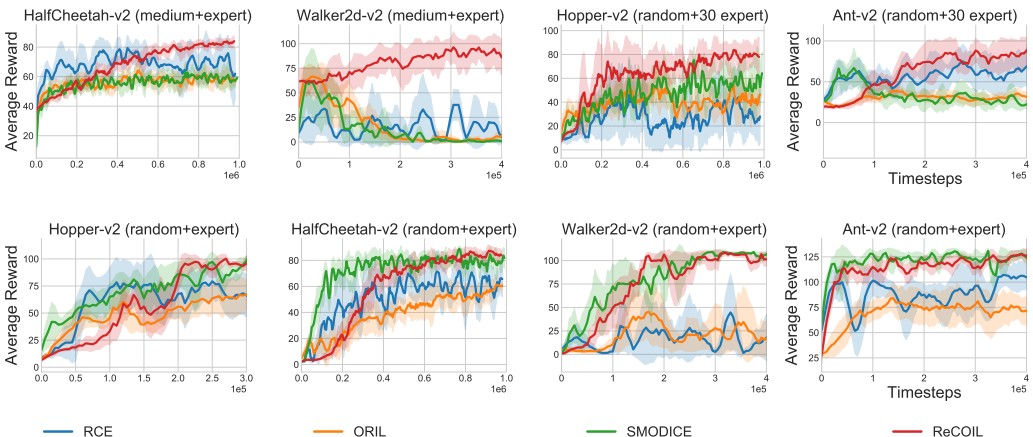

Figure 7: ReCOIL performs competitively in the setting of learning to imitate from diverse offline data. The results are averaged over 5 seeds

