# OpenReview forum: "Imitation from Arbitrary Experience: A Dual Unification of Reinforcement and Imitation Learning Methods"
_ICLR.cc/2023/Workshop/RRL — RRL 2023 Poster_

### Official Review · Reviewer_857Q · 2023-02-22
**Interesting framework, irrelevant to workshop theme**

**Rating:** 2
**Confidence:** 3

**Review:**

The authors propose a unifying framework for reinforcement learning based on the dual form of seeing RL as a convex problem with linear constraints. They also use this framework to propose a new algorithm for imitation learning from arbitrary data that relaxes the assumption from previous work that requires expert coverage of the problem space. The authors are thorough in their framework and method. While the empirical results of their proposed method lacks analysis, the major contribution of this work is in their theoretical framework.

However, the paper is only vaguely related to reusing prior computation for RL, in that the proposed method is for imitation learning with offline datasets. For this reason I rate the submission lower than the quality of the paper would suggest.

---

### Official Review · Reviewer_b2XH · 2023-02-25
**Generally solid paper framing RL and IL via the dual formation, but lacking slightly in story and experiments**

**Rating:** 2
**Confidence:** 3

**Review:**

This is a generally solid paper using the dual formation to provide some unified view of RL and IL algorithms. The paper provides some theoretical theorems, but lacks a good story that strings these theorems together. Experiments are descent in demonstrating the efficacy of their new algorithm. However, there is only one set of experiments provided. The contributions from this paper have good intentions, but it likely to only be of significance to a small portion of the community.